# Flow Experiences and Virtual Tourism: The Role of Technological Acceptance and Technological Readiness

**Chenyujing Yang [1], Shaocong Yan [1], Jingyu Wang [2] and Yongji Xue [1,*]**

1   School of Economics and Management, Beijing Forestry University, Beijing 100083, China;
    yangchenyujing@163.com (C.Y.); dacong0225@163.com (S.Y.)
2   School of Economics and Management, Zhejiang A&F University, Hangzhou 311300, China;
    wangjy@zafu.edu.cn
*   Correspondence: xyjbjfu@bjfu.edu.cn

**Abstract:** Virtual technology has brought new development opportunities to the tourism market and is expected to help the tourism industry cope with the challenges issuing from the COVID-19 pandemic. Given this context, in this study, we propose and test a model based on the SOR architecture, which includes tourists' experience of virtual tourism, technical readiness (TR), technical acceptance (TA), and tourists' virtual tourism intentions and the variables of flow experience, technical optimism, technical discomfort, perceived usefulness, perceived ease of use, adoption intention, and consumption intention. To this end, data were collected through a questionnaire survey of Chinese tourists ($n$ = 542). Then, we used a structural equation model (SEM) to test the hypothetical relationships between potential variables. The results showed that the flow experience delivered by the virtual tourism experience affects tourists' tendencies to use and intentions to consume virtual tourism. Second, flow experiences can make tourists more optimistic about virtual tourism technology, reduce tourists' technical discomfort, and enhance tourists' perceptions of usefulness and ease of use. We also found that tourists' intentions to use virtual tourism technology affect their intentions to travel on the spot. These findings provide useful insights for tourism practitioners, suggest new ideas for marketing and sustainable development in the virtual tourism industry, and verify the application of the integrated SOR and TAM framework in the field of tourism consumption.

**Keywords:** virtual tourism; flow experience; technical readiness; perceived usefulness; perceived ease of use; adoption intention; structural equation model; Chinese tourists

## 1. Introduction

The coronavirus disease 2019 (COVID-19) pandemic has had a significant impact on global economic and social development, particularly in the tourism industry. In the early stages of the pandemic, the suppression of coronavirus transmission was mainly based on non-pharmacological interventions (NPIs) [1]. During the pandemic, traditional infectious disease prevention methods were adopted, such as case detection, isolation, and personal protection (wearing masks), and new methods were also adopted, such as social distancing and travel restrictions [2]. Travel restrictions included border closures, access restrictions, and traffic control [3], and their role in infection prevention has been confirmed. Studies have shown that Australia's travel restrictions reduced COVID-19 cases and deaths by about 87% [4]. If China does not adopt a travel ban and a containment strategy, the number of COVID-19 cases is estimated to increase 67-fold [5]. However, travel restrictions also have certain limitations, some of which have a significant impact on the tourism economy. The United Nations World Tourism Organization (UNWTO) released a report stating that 2020 was 'the worst year in the history of tourism'. Affected by the pandemic, the number of international tourists decreased by one billion people that year, a decline of about 74% [6]. According to the data released by the Ministry of Culture and Tourism of China, the number

of domestic tourists in 2020 decreased by 52.1% and domestic tourism income decreased by 61.1% [7].

From another point of view, the impact of this major health incident on traditional tourism indirectly promoted the development of virtual technologies, such as cloud tourism and virtual scenic spots. Virtual technology has brought new development opportunities for the tourism market and is expected to help the tourism industry to cope with the challenges posed by COVID-19. Virtual tourism technology brings consumers a real sense of experience and immersion and is widely used in the panoramic experience of attractions [8]. It is a new choice for consumers who cannot travel in person. Furthermore, virtual tourism technology provides sensory information related to the destination for potential tourists and helps consumers to carry out practical tourism actions through the 'advance' experience of virtual tourism [9,10]. Due to the travel restrictions imposed as a result of the pandemic, virtual tourism has received more attention from consumers.

It is well known that there is a vague boundary between sustainable and unsustainable tourism [11]. However, it is undeniable that the development of traditional tourism does have a certain regional impact which may render tourist destinations vulnerable and have a negative impact on their cultural, social, economic, or environmental systems [12]. Sustainable tourism may provide a key perspective to reduce the impact of traditional tourism on tourist destination vulnerability. Specifically, the purpose of sustainable tourism is to ensure the well-being of future generations while meeting the current needs of tourists and tourism [13,14]. We believe that virtual tourism technology is an effective way to promote the sustainable development of the tourism industry in the context of the pandemic. On the one hand, in terms of environmental friendliness, the use of virtual tourism technology reduces the environmental pollution load of tourist destinations, reduces the pressure on their ecosystems caused by human activities, and may help the tourism industry to reduce carbon emissions. On the other hand, technological development often drives the development of tourism [15]. Technology provides consumers with more choices and makes it possible for tourists to change their consumption patterns, especially through the comparison of consumer costs. For example, information and communication technologies often imply new opportunities to help consumers quickly identify the 'best' accommodations, the 'best' restaurants, and the 'most popular' attractions [16]. Similarly, virtual tourism technology gives consumers the sensory experience of tourist destinations. Consumers can use this technology to make tourism decisions on the basis of such pre-experience, reducing the cost of tourism choices. In short, virtual technology may help sustainable tourism development by promoting innovative tourism development and reducing consumer travel costs.

To explore consumer attitudes and intentions to adopt virtual tourism technology, in this study, we take stimulus–organism–response (SOR) as the theoretical framework, combine Technology Acceptance Model (TAM) theory and Technology Readiness (TR) theory in the field of consumer behavior research, and take consumers as the research subjects (consumers with knowledge of virtual tourism technology and related activities) to construct the influencing factors on virtual tourism adoption intentions based on consumer flow experience. The purpose of this study is dual. First, we explore the influence of consumer flow experience on virtual tourism adoption intentions from the perspective of technology readiness and technology acceptance, and this work provides theoretical support and a reference for promoting the healthy, sustainable, and stable development of China's virtual tourism industry. Second, this work provides empirical evidence and insights for developers of virtual tourism technology, helping them to understand consumers' attitudes towards and perceptions of it and the ways in which consumers' flow perceptions affect their intentions to adopt virtual tourism technology.

## 2. Literature Review

### 2.1. Virtual Tourism

Perry and Williams first proposed virtual tourism, believing that it allows participants to experience simulations of real and non-real scenes and that it constitutes a new form of

business generated by the combination of virtual reality technology and tourism [17]. There is no consensus about a universally accepted definition of 'virtual tourism' as definitions of 'virtual reality' are multiple and often discordant [18]. In this paper, our understanding of virtual tourism is relatively broad. We believe that virtual tourism is a form of tourism based on real tourism landscape, using advanced technology to construct virtual environment (VE) to stimulate sensory experience. Simply speaking, virtual tourism is a form of tourism that uses virtual technology to achieve immersive sensory stimulation.

Virtual tourism is a sustainable environmental protection technology which may help to promote sustainable tourism by reducing unnecessary greenhouse gas emissions in transportation and improving 'virtual accessibility' [19]. In fact, virtual tourism technology had been developed to some extent before the outbreak began. Many theme parks, cultural heritage, and other tourist attractions had already introduced digital technology (AR, AI, etc.) to provide people with immersive experiences beyond time and space. For example, the Louvre, the Palace Museum, and the Smithsonian National Museum of Natural History have launched online virtual tour services for tourists. Virtual reality technology has the characteristics of visualization, immersion, and interaction [20], which can quickly reproduce real tourist attractions and bring about a revolution in tourism experience [21]. Existing studies on virtual tourism experience mainly focus on sensory enjoyment [22,23] and emotional experience [24].

At present, the research on virtual tourism mainly focuses on the definition of concepts [25], the realization of virtual technology [26], the advantages and disadvantages of virtual tourism, and destination marketing based on virtual tourism [27]. There are relatively few studies on virtual tourism from the perspective of tourists' experiences.

### 2.2. Theory of the SOR Model

The SOR model was originally developed as a concept in the field of psychology and is mainly used to explain the influence of environmental characteristics on users' psychological activities and behaviors. S denotes the external stimuli that affect the body (stimulus), O denotes the cognition of the organism (organism), and R denotes the response of the subject after receiving the stimuli through some changes in emotion or perception (response). Most SOR models are used to study consumer behavior. The SOR model has also been widely used in research on Chinese consumers, such as in research on Chinese rural consumer behavior [28], research on Chinese residents' purchase of energy-saving products [29], research on Chinese consumers' information avoidance behavior in the pandemic situation [30], and research on natural tourism participation behavior [31].

In fact, the SOR model is also widely used in tourism. Hew J. investigated mobile social tourism (MST) shopping among domestic Malaysian tourists based on the SOR framework, arguing that environmental stimuli directly or indirectly affect tourists' MST shopping intentions through intrinsic organism changes [32]. Su constructed a SOR framework to study environmental responsibility behavior in tourism using the eco-friendly reputation perception of a destination as the stimulus, consumption emotion as the organism, and tourism satisfaction and tourists' environmental responsibility behavior as the response [33]. In general, in the context of tourism consumption, external stimulation includes not only objective factors, such as tourism landscape, but also subjective factors, such as various services provided by tourist destinations and tourism reputation [34], which will affect tourists' perceptions and stimulate tourists to produce corresponding behavioral responses. We believe that the flow experience delivered by virtual tourism is similar to the service value provided by tourist destinations, which can be used as a stimulus to affect consumers' emotions and then affect their behavior.

### 2.3. Theory of Technology Readiness and the Technology Acceptance Model

The Technology Acceptance Model (TAM) is based on rational behavior theory. In this model, the user's final adoption behavior with respect to a certain technology is determined by the user's behavioral attitude. The so-called attitude comprises the user's concepts of

the good and evil of technology, including perceived usefulness and perceived ease of use. The TAM model has universal applicability and has been verified in different environments, such as mobile technology [35], virtual communities [36], and online games [37]. Similarly, the Technology Acceptance theory has been widely employed in the study of tourism-related behavioral intentions. It is believed that perceived usefulness and perceived ease of use are important factors affecting consumers' online travel reservation intentions [38], tourism app use [39], and tourism website use [40].

Technology Readiness (TR), which was proposed by the American scholar Parasuraman, refers to the tendency of people to accept and use new technologies to achieve the goals of their daily life or work [41]. Parasuraman believed that technology readiness has four dimensions: optimism, innovativeness, discomfort, and insecurity [41]. Optimism and innovativeness are the driving factors increasing technology readiness, while discomfort and insecurity are inhibitory factors. The relative advantages of the two factors determine individual adoption tendencies.

Many scholars combined TR with other models [42], such as the Theory of Planned Behavior, the Expected Confirmation Model, and the SST Attribute Model [43,44]. Similarly, a combination of TR and the TAM model has also been developed. Lin took the online stock trading system as an example to integrate TR into the TAM and developed it into a Technology Readiness Acceptance Model (TRAM). The results show that TR has a positive effect on perceived ease of use (PEU) and perceived usefulness (PU), and the effect on the use intention (UI) of online stock transactions needs to be mediated by PU and PEU [45]. Some scholars integrated TR into an extended TAM model, further verifying that TR has a positive impact on PEU and PU and confirming that TR has a positive effect on attitude [46].

## 3. Research Models and Assumptions

### 3.1. Flow Experience

In recent years, the concept of flow has attracted attention from scholars and practitioners. In 1975, Csikszentmihalyi proposed 'flow' as a positive psychological approach to understanding the best experiences [47]. Studies have shown that the flow experience affects participants' information acceptance, leading to changes in attitude and behavior [48], and that it can also affect consumers' repurchase intentions with respect to perceived value [49]. In the field of tourism and leisure, numerous studies have explored the effect of 'flow' on consumer behavior [50,51]. Studies have shown that 'flow' may be an important factor affecting consumer behavior and experience assessments in leisure environments (online or offline) [52], which may awaken emotions in experience and actively contribute to creating positive experiences [53,54]. Studies on nature-based tourism on Jeju Island, South Korea, showed that the flow experience is significantly positively correlated with satisfaction, environmental responsibility behavior, and destination loyalty [55]. In short, the flow experience is considered to be an important factor in awakening consumer behavior.

In recent years, scholars have also been concerned about the role of flow experience in virtual technology adoption and virtual platform use. As a marketing tool, virtual reality may increase the positive emotions of participants [56]. Virtual activities may enhance participants' sense of participation and 'flow experience' [57]. Research based on the SOR framework found that the quality of an online travel agency website has a significant influence on the flow experience, thereby affecting customer satisfaction and purchase intentions [58]. Based on this, we suggest that sensory stimulation brought by virtual tourism may awaken participants' emotions, give them a certain flow experience, and then affect their intention to use virtual tourism platforms. Considering these findings, we propose the following hypothesis.

**Hypothesis 1 (H1).** *Flow experience positively affects the intention of virtual tourism technology adoption.*

We also believe that the flow experience of using virtual tourism platforms affects participants' Technology Readiness (TR). It should be noted that TR is an overall psychological state rather than a measure of technical ability [59]. We believe that the flow experience of virtual tourism awakens the participants' mood, which may change the psychological state and bring changes in technology readiness. Considering these findings, we propose the following hypotheses.

**Hypothesis 2 (H2).** *Flow experience affects virtual tourism technology readiness.*

**Hypothesis 2a (H2a).** *Flow experience affects virtual tourism technology optimism.*

**Hypothesis 2b (H2b).** *Flow experience affects virtual tourism technology discomfort.*

*3.2. Technology Readiness and Technology Acceptance*

In order to understand the mechanisms of tourist intention formation in the context of virtual tourism, we take technology acceptance and technology readiness as influencing factors to predict the adoption of virtual tourism technology. The theoretical framework of the TAM helps us to understand the intention of travelers to use information technology in travel decision making [60]. TR evaluates people's emotional beliefs about new technology products or services on both positive and negative sides. In this study, we combine the TAM and TR to construct a simplified TRAM model.

In most studies, it is believed that the four dimensions of TR are relatively independent, representing different meanings and psychological processes [61]. However, in recent years, there have been studies that have taken different approaches, suggesting differences in the application of technology readiness in different fields. For example, studies by Ismail and others found that innovative spirit has a significant correlation with optimism and is not related to the other two constructs [62]. Taylor verified the applicability of TR in the insurance industry and the results supported only the validity of optimism and innovation [63]. On this basis, some scholars have proposed that the total score of four dimensions may not be optimal for predicting customer behavior. It is more persuasive and practical to separately discuss the impact of each dimension on the outcome variable [42]. The authors of this study believed that TR was intended to measure people's positive and negative emotional beliefs about new technology products or services, so it was necessary to evaluate users' customer satisfaction from two perspectives: positive feedback and negative feedback. Combined with the characteristics of virtual tourism technology, optimism and technical discomfort were finally selected to measure TR. Optimism is a positive attitude toward and belief in technology; that is, people believe that technology can provide more flexibility. Discomfort is a lack of perception of technology and a sense of pressure from technology; that is, individuals believe that technology is too complex, rather than being designed for ordinary people. According to Walczuch's research, there are differences in the impacts of the dimensions of TR on PE and PU. Optimism and insecurity have an impact on PU and PE, but innovation and discomfort are not strong [64].

The simplified TRAM model that we constructed for this study includes two parts. The first part is the impact of TR on PU and PUE. The specific research hypotheses are as follows:

**Hypothesis 3 (H3).** *Technology readiness affects technology acceptance.*

**Hypothesis 3 (H3a).** *Technology optimism positively affects perceived usefulness of technology.*

**Hypothesis 3 (H3b).** *Technology optimism positively affects perceived ease of use.*

**Hypothesis 3 (H3c).** *Technology discomfort negatively affects perceived usefulness.*

**Hypothesis 3 (H3d).** *Technology discomfort negatively affects perceived ease of use.*

The second part of our simplified TRAM model examines the applicability of the TAM in virtual tourism. In other words, we believe that consumers' positive perceived usefulness and perceived ease of use will positively affect their intention to adopt virtual tourism technology, and their perceived ease of use will also have an impact on perceived usefulness. Therefore, we propose the following hypotheses:

**Hypothesis 4 (H4).** *Technology acceptance positively affects virtual tourism technology adoption intentions.*

**Hypothesis 4 (H4a).** *Perceived usefulness of technology affects virtual tourism technology adoption intentions.*

**Hypothesis 4 (H4b).** *Perceived ease of use of technology affects virtual tourism technology adoption intentions.*

**Hypothesis 4 (H4c).** *Perceived ease of use of technology affects perceived usefulness of technology.*

### 3.3. Adoption Intentions and Consumption Intentions

Virtual reality experiences have a positive and significant impact on destination behavior intentions [65,66]. Advertising through VR can greatly increase the possibility of future visits [67]. Based on this, we propose the following hypothesis:

**Hypothesis 5 (H5).** *Intentions to adopt virtual tourism technology positively affect intentions to travel on the spot.*

Overall, the proposed model is shown in Figure 1. Based on the SOR framework, we integrate the TAM model and the TR model to explore the formation mechanism of virtual tourism consumption intentions. Regarding the model figure, there are two points to explain. Firstly, the SOR framework can better solve the problem of research bias caused by consumer unfamiliarity. Virtual tourism technology is relatively novel for consumers, which also means that it is unfamiliar. If the TR or TAM model is used without intuitive feelings (stimulus), there may be relatively large errors. On the contrary, if consumers are given external stimuli, it may be more rigorous and accurate to examine their intentions to adopt virtual tourism technology on the basis of experiencing sensory stimuli from virtual tourism, which is exactly what the SOR model enables. Secondly, a single TR or TAM model is difficult to apply. Consumers have various emotions and perceptions in the face of virtual tourism technology, even when there are links between emotions and perceptions. At the same time, the concept of the 'organic' in the SOR model is extensive, and embedding the TR and TAM models may effectively improve the scientific research, which is one of the main research focus of this paper.

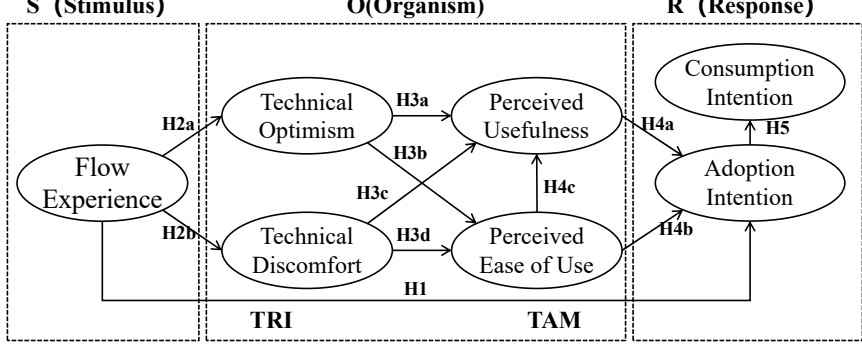

**Figure 1.** The proposed model of SOR.

## 4. Research Methods

In the present study, we adopted quantitative methods and empirical model hypothesis testing, which made it necessary to collect and analyze relevant data. We collected samples by issuing questionnaires online. Firstly, the widely used mature scale was employed to ensure the credibility of the study. Each construct item was measured with a seven-level Likert scale, from level 1 (deeply disagree) to level 7 (deeply agree).

The survey was conducted from January to February 2022 and the respondents were randomly selected consumers from Beijing, Guangzhou, Shanghai, and Jiangsu Provinces in China. Studies have shown that people may prefer to visit natural scenic spots during the COVID-19 pandemic [68]. Therefore, we selected natural scenic spots in the Panoramic Virtual Tourism Network, the largest virtual e-commerce tourism platform in China, for our study. The Panoramic Virtual Tourism Network (Panoramic) uses novel 720-degree, three-dimensional panoramic technology to bring users immersive experiences with three-dimensional and high-definition effects. As of 2018, the Panoramic network includes more than 400 cities in China and abroad, offering high-definition, 720-degree, 3D panoramic views and 3D virtual roaming of more than 10,000 scenic spots. In our survey, we inserted URL links to require respondents to experience virtual tourism and help stimulate their flow experiences.

A total of 680 questionnaires were collected. We posed the test question, 'What are the scenic spots of virtual tourism experience in the link?' to confirm whether the respondents actually followed the link and experienced the scenes. Then, 542 valid questionnaires were obtained after the preliminary screening of the questionnaire responses for integrity and data reliability, giving an effective response rate of 79.71%. Gefen pointed out that the appropriate minimum sample size for structural equation modeling in management information system research is 200 participants [69]. Therefore, we believe that the sample size for this study is acceptable.

Most of the respondents were between 26 and 35 years old, accounting for more than 60%. In terms of gender distribution, the gender gap of the respondents was relatively small. Regarding level of education, most participants had received higher education, generally college or undergraduate education. The health status of most respondents was considered healthy. Regarding the use of virtual tourism platforms, 52.40% of respondents used virtual tourism platforms, of which 31.34% were used before tourism experiences and 22.56% were used during tourism experiences. The collected sample characteristics are shown in Table 1.

**Table 1.** Characteristics of the survey respondents.

| Sociodemographic Variable | Absolute Frequency | Percentage |
|---|---|---|
| Gender | | |
| Male | 283 | 52.21% |
| Female | 259 | 47.79% |
| Age group | | |
| ≤25 years old | 97 | 17.90% |
| Between 26 and 35 years old | 338 | 62.36% |
| ≥35 years old | 107 | 19.74% |
| Area | | |
| Jiangsu Province | 95 | 17.53% |
| Zhejiang Province | 75 | 13.84% |
| Beijing | 88 | 16.24% |
| Shanghai | 81 | 14.94% |
| Guangdong Province | 203 | 37.45% |

**Table 1.** *Cont.*

| Sociodemographic Variable | Absolute Frequency | Percentage |
|:---:|:---:|:---:|
| Health | | |
| Health | 314 | 57.93% |
| General | 167 | 30.81% |
| Unhealthy | 61 | 11.25% |
| Monthly Income (RMB) | | |
| Rather not say | 19 | 3.51% |
| <2000 | 9 | 1.66% |
| 2001–4000 | 29 | 5.35% |
| 4001–6000 | 72 | 13.28% |
| 6001–8000 | 131 | 24.17% |
| 8001–10,000 | 128 | 23.62% |
| >10,001 | 155 | 28.60% |

## 5. Data Analysis

The measurement items used in this paper were revised according to the existing mature scale. The measurement of 'flow experience' was combined with Yao Yanbo's research on social media and tourism intentions [70]. Based on the existing research, the technology readiness system was improved according to the characteristics of virtual tourism [71]. The items of perceived usefulness and perceived ease of use referred to the research design of Davis; at the same time, they referred to the adjustment of scholars' research on experiential tourism in the new COVID-19 pandemic [72,73].

The model constructed in this study has a high level of complexity, consisting of seven variables. Therefore, partial least squares (PLS) analysis was used to handle this complexity, as this method is suitable for overcoming the abnormal distribution of data. Before the data analysis, we used the SPSS 26.0 software to test the reliability and common method deviation of the data. Then, the AMOS 21.0 software (IBM, New York, NY, USA) was used for the data analysis. The analysis steps of the software included two stages, namely, a measurement model and a structural model.

### 5.1. Reliability and Common Method Deviation

Harman single-factor tests showed that the variance interpretation rate of the first polymerization factor without rotation was 28.38%, less than 50%, and the factor with a characteristic root greater than 1 was more than one. The common method latent factor (CMV) test showed that the model was not significantly improved after adding the common method latent factor ($\Delta x^2/df = 0.151$; $\Delta CFI = 0.017 < 0.1$; $\Delta TLI = 0.012 < 0.1$; $\Delta RMSEA = 0.005 < 0.05$), so the common method bias in the measurement was acceptable for the bias risk of this study.

SPSS 26.0 software was used for the reliability testing of the data. The results showed that the overall Cronbach's alpha index of the questionnaire was 0.868, indicating adequate stability and reliability. The non-standardized estimates of each variable reached a significant level, and the standardized factor load was generally close to or greater than 0.7. At the same time, the CR values of each variable were greater than 0.6, indicating that the compositional reliability was relatively suitable and had sufficient internal consistency.

### 5.2. Measurement Model

The first step of the measurement model was to ensure that the reliability and validity of the items fit the criteria of convergent validity and discriminant validity. In this study, we found that the value of each loading factor was greater than 0.6, which we considered to be standard. From the AVE value, it is generally believed that the extracted average variance (AVE) should be higher than 0.5, but we could accept 0.4. This is because Fornell and Larcker proposed a structure with an AVE that was less than 0.5 when the comprehensive reliability

was higher than 0.6, and the effectiveness of the convergence was still sufficient [74]. In general, we believe that the data collected are reliable and can be used for further research.

Second, we determined whether there was sufficient discriminant validity between variables. The value of the square root of the AVE represented the aggregation of factors, and the correlation coefficient represented the correlation. If the aggregation of factors is higher than the correlation, this indicates that the discriminant validity of factors is adequate. We found that the data in this study met this requirement. In sum, as shown in Tables 2 and 3, we believe that this set of data has sufficient reliability and convergence validity.

**Table 2.** Results of the measurement model.

| Construct | Measurement Items | Factor Loading | Cronbach's Alpha | CR | AVE |
|---|---|---|---|---|---|
| Flow Experience | FE1 | 0.757 | 0.732 | 0.834 | 0.556 |
| | FE2 | 0.727 | | | |
| | FE3 | 0.757 | | | |
| | FE4 | 0.741 | | | |
| Technical Optimism | TO1 | 0.746 | 0.719 | 0.817 | 0.473 |
| | TO2 | 0.714 | | | |
| | TO3 | 0.581 | | | |
| | TO4 | 0.717 | | | |
| | TO5 | 0.670 | | | |
| Technical discomfort | TD1 | 0.850 | 0.896 | 0.923 | 0.707 |
| | TD2 | 0.844 | | | |
| | TD3 | 0.846 | | | |
| | TD4 | 0.830 | | | |
| | TD5 | 0.834 | | | |
| Perceived usefulness | PU1 | 0.719 | 0.712 | 0.823 | 0.538 |
| | PU2 | 0.715 | | | |
| | PU3 | 0.744 | | | |
| | PU4 | 0.754 | | | |
| Perceived ease of use | PEU1 | 0.682 | 0.654 | 0.794 | 0.492 |
| | PEU2 | 0.718 | | | |
| | PEU3 | 0.686 | | | |
| | PEU4 | 0.717 | | | |
| Willingness to adopt | AI1 | 0.788 | 0.665 | 0.818 | 0.599 |
| | AI2 | 0.740 | | | |
| | AI3 | 0.793 | | | |
| Willingness to travel | CI1 | 0.798 | 0.647 | 0.810 | 0.587 |
| | CI2 | 0.750 | | | |
| | CI3 | 0.749 | | | |

**Table 3.** Correlation matrix.

| | FE | PU | PUE | AI | CI | TO | TD |
|---|---|---|---|---|---|---|---|
| FE | **0.746** | | | | | | |
| PU | 0.593 | **0.733** | | | | | |
| PUE | 0.514 | 0.671 | **0.701** | | | | |
| AI | 0.65 | 0.721 | 0.576 | **0.774** | | | |
| CI | 0.587 | 0.652 | 0.606 | 0.630 | **0.766** | | |
| TO | 0.634 | 0.714 | 0.619 | 0.677 | 0.617 | **0.688** | |
| TD | −0.105 | −0.132 | −0.128 | −0.092 | −0.163 | −0.095 | **0.841** |

Note: Diagonal elements in bold show the square root of AVE.

### 5.3. Structural Model

The two test steps described above show that the survey measurements were effective and reliable and that the existing assumptions could be assessed in the next phase.

The measurement model tested the impact of consumer flow experience, technology acceptance, technology readiness, and technology adoption. The empirical results (Table 4) showed, firstly, that a user's flow experience had a significant positive impact on their intention to use a virtual tourism platform ($\beta = 0.221$, $p < 0.01$), suggesting that H1 is true. Second, a user's flow experience had a significant impact on their perceived technology readiness. The impact coefficients of technology optimism and technology discomfort were 0.890 and $-0.120$, respectively, at the 1% and 5% levels, suggesting that H2a and H2b hold. Third, technology readiness had a significant impact on consumer acceptance of technology. The impact coefficients of technology optimism on perceived usefulness and perceived ease of use were 0.457 and 0.916, respectively, and both were significant at the 5% level, i.e., suggesting that H3c and H3d are tenable. However, it should be noted that the impact of technological discomfort on perceived usefulness and perceived ease of use was not significant, suggesting that H3a and H3b are untenable. Fourth, the TAM was generally applicable in the field of virtual tourism technology adoption. Perceived usefulness had a positive impact on consumers' intentions to adopt virtual tourism technology ($\beta = 0.935$, $p < 0.05$), while perceived ease of use had no direct impact, but it could positively affect perceived usefulness ($\beta = 0.550$, $p < 0.01$) and thus affect the intention to adopt the technology. Fifth, the intention to use virtual tourism technology could improve the intention of consumers to travel on the spot ($\beta = 0.936$, $p < 0.01$).

**Table 4.** Summary of the results of structural equation modeling.

| Hypothesis | Influence Path | Estimate | S.E. | C.R. |
|---|---|---|---|---|
| H1 | Adoption Intention←Flow Experience | 0.221 ** | 0.098 | 2.219 |
| H2a | Technical Optimism←Flow Experience | 0.890 *** | 0.073 | 12.528 |
| H2b | Technical Discomfort←Flow Experience | −0.120 ** | 0.090 | −2.248 |
| H3a | Perceived Usefulness←Technical Discomfort | −0.013 | 0.015 | −0.430 |
| H3b | Perceived Ease of Use←Technical Discomfort | −0.057 | 0.018 | −1.307 |
| H3c | Perceived Usefulness←Technical Optimism | 0.457 ** | 0.164 | 2.369 |
| H3d | Perceived Ease of Use←Technical Optimism | 0.916 *** | 0.066 | 9.232 |
| H4a | Adoption Intention←Perceived Usefulness | 0.935 ** | 0.530 | 2.000 |
| H4b | Adoption Intention←Perceived Ease of Use | −0.138 | 0.625 | −0.318 |
| H4c | Perceived Usefulness←Perceived Ease of Use | 0.550 *** | 0.258 | 2.706 |
| H5 | Consumption Intention←Adoption Intention | 0.936 *** | 0.063 | 11.505 |
| Goodness of fit indexes | CMIN/DF = 1.414 ($p = 0.000$) GFI = 0.913; AGFI = 0.896 RMSEA = 0.038 | | | |

Note: ** and *** denote statistical significance at 5%, and 1%

### 5.4. Test of Mediating Effects

Based on the above analysis, we believe that technology acceptance and technology readiness will mediate the influence of flow experience on virtual tourism technology adoption intentions. Therefore, controlling for gender, age, and income, we conducted a bootstrap sampling regression analysis, with technology optimism as the mediating variable. It should be noted that in the previous stage of research, we found that the impact of technology discomfort on perceived ease of use and perceived usefulness was not significant. Therefore, in this step, we mainly analyzed the mediating effect related to technology optimism.

The results of the bootstrap analysis showed that the total mediating effect of technology optimism, perceived usefulness, and perceived ease of use was 0.394, and the confidence interval was [0.308, 0.492], without 0. Furthermore, we conducted a further mediating effect test to explore the mediating effect and the difference between the effects of technology readiness and technology acceptance. The results show that the mediating effect

could be reflected by technology optimism and perceived usefulness, and the proportions of the two effects were 19.53% and 9.70%, respectively. Technology optimism and perceived usefulness could also mediate the influence of flow experience on technology adoption intention. The mediating effect value was 0.097, and the confidence interval of the effect proportion was 14.49%, or [0.062, 0.139], excluding 0. The mediating effect of perceived ease of use was not obvious, and it only existed with technology optimism and perceived usefulness. The chain mediating effect value was 0.039, accounting for 5.85% of the total mediating effect. Overall mediation test results are shown in Table 5.

**Table 5.** Summary of direct, indirect, and total effects.

| Path | Effect | 95% Confidence Interval | | | Establish |
|---|---|---|---|---|---|
| | | BootSE | BootLLCI | BootULCI | |
| Total effect of FE on AI | | | | | |
| FE→AI | 0.672 | 0.035 | 0.605 | 0.740 | Yes |
| Direct effect of FE on AI | | | | | |
| FE→AI | 0.278 | 0.037 | 0.205 | 0.351 | Yes |
| Indirect effects of FE on AI | | | | | |
| Ind1: FE→TO→AI | 0.131 | 0.037 | 0.062 | 0.209 | Yes |
| Ind2: FE→PE→AI | 0.014 | 0.011 | −0.003 | 0.038 | No |
| Ind3: FE→PU→AI | 0.065 | 0.020 | 0.030 | 0.108 | Yes |
| Ind4: FE→TO→PE→AI | 0.020 | 0.013 | −0.006 | 0.046 | No |
| Ind5: FE→TO→PU→AI | 0.097 | 0.020 | 0.062 | 0.139 | Yes |
| Ind6: FE→PE→PU→AI | 0.027 | 0.010 | 0.011 | 0.049 | Yes |
| Ind7: FE→TO→PE→PU→AI | 0.039 | 0.010 | 0.024 | 0.06 | Yes |
| Total | 0.394 | 0.048 | 0.308 | 0.492 | Yes |

## 6. Discussion

From the perspective of user experience, based on the integrated model of the SOR framework and the TAM, we focused on the impact of tourists' flow experiences on virtual tourism technology adoption under the framework of technology readiness and technology acceptance in order to provide a reference for tourism practitioners and R&D personnel involved in the design of virtual tourism technology. According to the results of the empirical analysis, the TRAM constructed in this study was strongly supported overall, but some hypotheses did not hold. The empirical results of the model are shown in Figure 2.

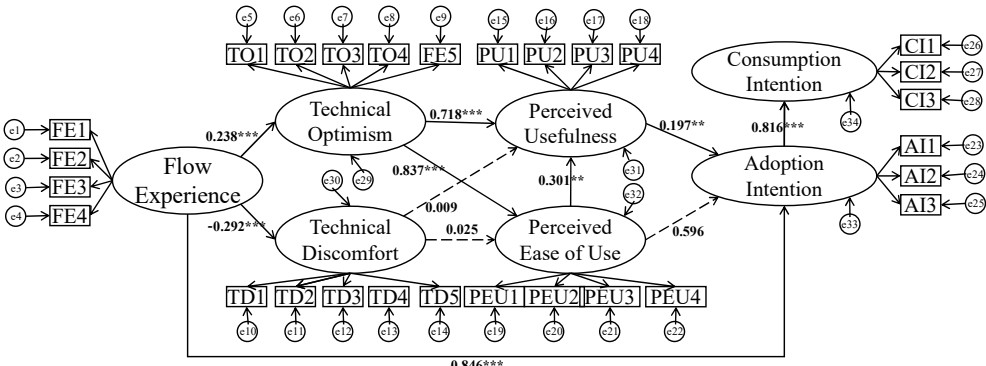

**Figure 2.** Empirical results of the structural equation model. (** and *** denote statistical significance at 5% and 1%.)

Flow experiences positively affect consumers' intentions to use virtual tourism technology and their consumption intentions. Consumers' flow experiences in virtual tourism give them opportunity to feel the new experiences delivered by virtual tourism more

intuitively. At the same time, this immersive experience not only deepens consumers' awareness of virtual tourism but also further awakens their inner interest in this experience. As is often seen in Chinese shopping malls, businesses try to generate purchase desire among consumers through free VR experience promotions, thereby promoting consumption. This study verifies the applicability of this promotion method in virtual tourism. For consumers, flow experiences arouse their curiosity in and meet their expectations for virtual tourism technology and increase their recognition and satisfaction. This positive perception promotes consumers' intentions to use virtual tourism technology and their consumption desire.

Consumer flow experiences can affect the acceptance of virtual tourism technology through the readiness for virtual tourism technology, thereby affecting intentions to use virtual tourism technology. First, the flow experiences of consumers significantly affect their perceptions of virtual tourism technology. Specifically, the flow experiences of consumers will stimulate their inner positive emotions, which can significantly improve their recognition of and optimism toward virtual tourism, so consumers can have higher expectations for virtual tourism technology. At the same time, positive flow experiences can eliminate consumers' unfamiliarity with virtual tourism technology, reduce their concerns, enhance their confidence, and reduce their discomfort with virtual tourism technology, as people often have an instinctive psychology of avoidance in the face of a new, strange concept. Flow experiences in virtual tourism can improve the cognitive level of consumers to a certain extent and can reduce the technology discomfort caused by unfamiliarity. Second, virtual tourism technology readiness, especially optimism, has a significant positive impact on consumer technology acceptance. This is consistent with existing research conclusions [64]. The flow experience makes consumers behave more intuitively when evaluating the quality of virtual tourism technology. A positive experience will make consumers more optimistic. Evaluations of this form of tourism tend to be positive and hopeful. It is believed that it not only has a low threshold for use, but it can also bring more interesting tourism experiences and higher utility levels. Therefore, a sound sense of experience will make consumers more optimistic about virtual tourism technology, thereby affecting perceived ease of use and usefulness. At the same time, it is worth noting that another factor of technology readiness, technology discomfort (a negative emotion), has no significant inherent influence on technology acceptance (including perceived ease of use and usefulness). As Parasuraman pointed out, although positive and negative emotions exist at the same time and interact with each other in the individual, one must be dominant, and dominant emotions vary from person to person [41]. This shows that, for most consumers, discomfort with virtual tourism technology is relatively weak or that the intensity of this discomfort is still within the allowable range, meaning that it will not have a significant impact on the process of technology adoption.

In addition, perceived ease of use and perceived usefulness have a significant positive impact on consumers' intentions to adopt technology. On the one hand, the perceived ease of use of virtual tourism technology reduces the cost and threshold of participating in virtual tourism and directly contributes to the formation of consumption intentions. On the other hand, the more convenient and accessible consumers perceive a new technology or model of virtual tourism to be, the higher is its perceived usefulness [75] and the stronger their intention to adopt or consume virtual tourism technology. It can be seen that the ease of use of virtual tourism technology is prominent, so reducing its use threshold and increasing the degree of convenience should become the focus of tourism practitioners and technology R&D personnel. At the same time, perceived usefulness also has a significant positive impact on the intention to consume virtual tourism, so the improvement of the richness of virtual tourism experiences should also be considered.

In the process of generating the intention to use virtual tourism technology based on flow experience, technology readiness and technology acceptance play an inherent mediating effect, and there are significant differences. This is consistent with the above conclusion. Among the mediating effects, optimism in relation to technology readiness is

more prominent and it is also an inevitable link in the process of virtual tourism technology use intention formation. Therefore, we suggest that the experience and promotion of virtual tourism technology should be based on the driving factors, especially to enhance technology optimism, give consumers more freedom and flexibility to travel, and highlight the core value of virtual tourism technology. At the same time, the optimism toward and perceived usefulness of virtual tourism technology can mediate the influence of flow experiences on technology adoption intentions; that is, flow experiences can enhance the perception of the utility of virtual tourism technology by improving the travel expectations of consumers, stimulating, finally, the intention to use technology. In addition, technology optimism, perceived ease of use, and perceived usefulness can jointly play a chain mediating role, which provides a new choice for the path of the influence of flow experience. Therefore, we suggest that tourism practitioners and technology R&D personnel should systematically construct schemes for the development and operation of virtual tourism technology based on consumer experience levels. Starting from the flow experience, they should realize the progressive layers of flexibility, convenience, and utility, continuously accumulate the positive emotions in consumers with respect to use and promote the generation of technology use intentions and consumption desire.

The intention to use virtual tourism technology will positively promote real tourism intentions. In previous studies, a considerable number of scholars believed that there was an alternative relationship between virtual tourism and real tourism [8]. Even as technology advances, the former will gradually replace the latter, especially under the impact of major health events such as COVID-19. However, there is no doubt that virtual tourism technology promotes a positive change in attitude towards destinations by creating a 'presence' for potential tourists, resulting in a higher intention to visit sites [23]. With the COVID-19 pandemic in mind, we must recognize that, although pandemic prevention and control measures limit people's daily travel, they also add to the growing backlog of consumers' travel intentions. With the normalization of people's daily life in the post-pandemic era, this strong demand for real tourism will create another period of opportunity for tourism. The consumption of virtual tourism has given consumers more fantasies about destinations. As with attractive advertising for a film, it has led to consumers' strong intentions to travel on the spot. This directly supports the 'virtual–real linkage' of tourism.

Further, we believe that in the post-pandemic era virtual tourism will be sustainable. On the one hand, it provides a new idea for the traditional tourism industry, from an offline to an 'online and offline' development mode, which brings new impetus to innovation in the traditional tourism industry. On the other hand, the technology of virtual tourism is relatively mature, the technical threshold has been greatly reduced, and an effective business operation mode has been initially formed which has entered a stage of rapid development, laying a solid foundation for the wide spread of virtual tourism.

## 7. Contributions and Limitations

The above research results offer some contributions for the marketing and technological development of tourism and thus provide useful insights for tourism stakeholders. The following sections discuss these points as well as the limitations of this study and future research recommendations.

### 7.1. Theoretical Contribution

Proving the significant influence of flow experience on virtual tourism intention formation is the original contribution of this study. As mentioned in the literature review, the role of flow experience in virtual tourism has not been effectively confirmed and the relationship between the two structures has not been empirically solved. In this context, this survey explored the role of Chinese consumers' flow experiences in virtual tourism intention formation. The results reveal the attitudes of tourists toward virtual tourism against the backdrop of the COVID-19 pandemic and how these attitudes are affected by consumers' perceptions of technology. This broadens some boundaries for tourism

research and deepens the exploration of influencing factors regarding tourists' virtual tourism behavior in tourism research.

More specific theoretical contributions emerged from our interactive use of models. When exploring intentions to engage in virtual tourism, we adopted the SOR model and the TAM and we integrated the Technology Acceptance theory into them. We found that the combination of the Technology Acceptance Model and the SOR model had a better fitting effect. By exploring the effect of technology on consumer experience and behavior intentions, this paper provides a reference and ideas for the exploration of consumer behavior and intention formation in marketing studies.

### 7.2. Application Value

This research provides a new avenue for tourism practitioners to conduct marketing. In Chinese shopping malls, we can often find free VR experience activities, but there is no research exploring how this free experience promotes consumption. In this study, we investigated the influence of flow experience on intentions to adopt virtual tourism technology in the field of virtual tourism, and our findings offer new marketing ideas for tourism practitioners. This is expected to be a breakthrough that will broaden the marketing path of virtual tourism.

We found that perceived usefulness and perceived ease of use have significant impacts on the use of virtual tourism platforms, which are affected by the psychological states of users, particularly the degree of technology acceptance. Our research also provides a reference for the optimization of virtual tourism technology, suggesting that technology developers should pay more attention to users' experiences with their products to promote the purchase and use of their technology.

In addition, we found that the relationship between virtual tourism adoption intentions and field tourism consumption intentions is a positive one. This provides some new ideas for future research on the sustainable tourism industry. For example, people are concerned about how ICT enables tourists to participate in the protection of natural assets in an innovative way, which helps to make nature-based tourism more sustainable [11]. We try to provide some ideas from the perspective of virtual technology. We believe that the use of virtual technology is a good mode of innovation, which can help solve the problem of sustainable tourism based on nature. It should be noted that this is still a very complex problem.

### 7.3. Limitations and Future Research

Although there are clear contributions, the current research still has some limitations. The first drawback stems from the cross-sectional nature of the study, which limits the width of the range of the results of the model. Due to limitations of time and resources, we only used cross-sectional data from five regions in China; thus, our conclusions have room for further optimization. In future studies, panel data can be used for the construction and analysis of a more comprehensive model. Second, we adopted multi-variable decentralized research, and some variables' accuracies were still insufficient. In future research, further refinement of variables can be considered and more meaningful conclusions can be obtained through more detailed and sophisticated questionnaire designs. Third, in the discussion of technology acceptance, we only explored this factor from the perspective of technology optimism and technology discomfort and did not test the impacts of the four dimensions of technology acceptance on the intention to engage in virtual tourism, which may be the direction of our subsequent in-depth study. Fourthly, we only judged the relationship between virtual tourism intention and actual tourism and did not explore the specific path of transformation.

**Author Contributions:** Conceptualization, C.Y., S.Y. and J.W.; methodology, C.Y. and S.Y.; data collection, C.Y. and S.Y.; data analysis, C.Y.; writing—original draft preparation, C.Y. and S.Y.; writing—review and editing, C.Y., J.W. and Y.X.; visualization, C.Y., J.W. and S.Y.; supervision, C.Y. and Y.X.; project administration, C.Y. and Y.X. All authors have read and agreed to the published version of the manuscript.

**Funding:** This research was funded by the National Social Science Fund of China 'Research on the Driving Mechanism and Realization Path of Entrepreneurship for the Poor Based on Ecological Engineering Participation' (18BGL052).

**Data Availability Statement:** The data reported in this paper were collected by its authors through an online survey and are available to anyone upon request.

**Conflicts of Interest:** The authors declare no conflict of interest. The funding agency did not influence the data collection, analysis, interpretation, or the conclusion of this article.

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
