# Peer review of "Flow Experiences and Virtual Tourism: The Role of Technological Acceptance and Technological Readiness"

_sustainability, doi:10.3390/su14095361_

Round 1

Reviewer 1 Report

The article contains interesting research on virtual tourism, which during the COVID-19 pandemic was often a substitute for real travel. Many tourist attractions, such as museums, amusement parks, national parks, offered virtual tours using websites, you tube or Facebook. The authors used the theory of flow experience and combined models of Technology Acceptance and the SOR model to verify the hypotheses. The research carried out on a sample of over 600 respondents is methodologically correct, and statistical analyzes and modeling do not raise any objections. Proving the significant influence of the flow experience on virtual tourism intention is the original contribution of this study. The research results will undoubtedly be a practical inspiration for tourism marketing.

There has been a lot of work on virtual tourism in recent years, I suggest the authors to expand section 2.1. o adding the latest works in this field, displaying articles on the use of virtual tourism and modern technology in the marketing of the destination.

Author Response

Thank you very much for taking time to review this manuscript. I really appreciate your recognition of our research topics and methods ! 

You mentioned that we would like to expand the relevant research on virtual tourism in chapter 2.1 and add the latest works in this field. According to your opinion, 7 new articles have been added to show the application of virtual tourism and modern technology in destination marketing.

Reviewer 2 Report

The subject of this paper is very interesting and it would be fascinating to research in depth the relationship between virtual tourism and actual travel. It is not so clear whether intention to travel translates into actual travel, so this assumption is not always a useful one. However, the authors should be commended on their robust data collection and analysis. This cannot be faulted. My problem is that it is difficult to get a sense of where the contribution to knowledge and research lies here, because there are so many models, concepts and variables. The Technology Acceptance and Readiness hypotheses are rather basic (H3 and H4) so they do not really take this field in any specific direction. We do not learn too much about virtual tourism either. What is the nature of it and how does it connect to real experiences? I am not sure if it can even be described as tourism, rather as a virtual experience. I am assuming it only lasts for an hour or two? I am not sure that it was necessary to bring in the SOR model AND flow. Flow is a really complex concept and it is difficult to measure quantitatively. Here, I cannot really get a sense of what it means or how is was measured. I would suggest a simplification of this paper. Less is more sometimes. The authors aim to produce an almost purely quantitative piece of research, which works at that level, but some of the elements need more discussion and definition to be truly meaningful. If a member of the tourism industry or even a technology experience designer came across this paper, they would not be able to take much away from it to use for development. Of course, academic papers can exist without connection to the industry, but this is a practice that should perhaps be discouraged. The authors did an excellent job academically, there is no question, but my suggestion would be to go deeper into one or two issues and avoid skimming superficially over the others. The research will still be interesting, probably more so. The authors can probably form more than one paper from the same dataset in this case. But please define and discuss the constructs in more depth. 

Reviewer 4 Report

I think that the article is interesting and is worth being published. I would suggest making some revisions:

1 - I think it might be useful to explain better the phenomenon of SOR (paragraph 2.2). In particular, since SOR is a psychological concept that has been widely applied to the study of consumer behavior, it would be curious to know why you have decided to apply it to the study of virtual tourism.

2 - I think that you could analyze better Figure 1. This illustration contains much interesting information. I would suggest adding a paragraph with the description of the figure, following your paragraph lines 239-242.

3 - I think it is not reasonable to talk about the research results that are not present in the paper. I would suggest deleting the paragraph lines 386-389.

4 - It might be interesting to precise in paragraph 7.1 the concrete scholarly fields in which you position your paper (e.g. Tourism Studies, Museology, Art Economics, Urbanism, etc....). 

Round 2

Reviewer 2 Report

The authors made a clear effort to address my comments and to provide a response in the text too. I still think that there are too many constructs, concepts and variables in the paper but the research is robust and the subject is interesting.